# Investigation of a Simple Viscoelastic Model for a PVC-Gel Actuator under Combined Mechanical and Electrical Loading

**DOI:** 10.3390/ma16031183

**Published:** 2023-01-30

**Authors:** Maorong Zhang, Johnny Jakobsen, Ruiqin Li, Shaoping Bai

**Affiliations:** 1School of Mechanical Engineering, North University of China, Taiyuan 030051, China; 2Department of Materials and Production, Aalborg University, 9220 Aalborg, Denmark

**Keywords:** PVC gels, viscoelastic model, actuator, deformation, voltage

## Abstract

PVC gels are gaining more attention in the applications of soft actuators. While their characteristics have been extensively studied experimentally, precise models that predict the deformation due to imposed mechanical and electrical forces are not yet available. In this work, a viscoelastic model based on a combination of a Maxwell and a Kelvin–Voigt model is developed to describe the responsive deformation of the actuator. The model parameters are tuned using data obtained from a unique experimental setup. The PVC gel used in the actuator is made from PVC and dibutyl adipate (DBA) together with a tetrahydrofuran (THF) solvent. A full factorial test campaign with four and three levels for the mechanical and electrical forces, respectively, are considered. The results showed that some of the viscoelastic response could be captured by the model to some extent but, furthermore, the stiffness behavior of the PVC gel seemed to be load-type-dependent, meaning that the PVC-gel material changed stiffness due to the magnitude of the electrical force applied and this change was not equal to a similar change in mechanical force.

## 1. Introduction

Soft actuators, compared to rigid actuators, have advantages of being able to provide large actuation strain, high flexibility, and low noise, and they are lightweight, easy to process and fabricate, etc. One class of soft actuators is the electroactive polymers (EAP), which change size or shape upon electric stimulation; thus, they have been used in soft actuators and devices, for example, PVC-gel-based actuators [1,2]. PVC-gel-based soft actuators have demonstrated superior performance mimicking human muscles and may provide good stability and durability. Moreover, they show great potential for applications in soft-robotic and unconventional-actuator setups, generating impressive innovative momentum with ongoing improvements in the field of soft robotics and drive concepts [3,4]. Experimental studies on PVC gels to understand their material behavior and chemical structures have been reported [5,6,7]. However, few attempts on theoretical-model development for these materials have been made. It is essential to have accurate models when actuator control algorithms are to be formed.

PVC-gel actuators can take a very simple planar setup, which consists of a PVC film sandwiched between two compliant electrodes. When a voltage is applied, the attraction of the electrodes causes direct stress in the PVC gel. N. Ogawa et al. established a preliminary static model of a mesh-electrode-based PVC-gel actuator, which considered both the contraction strain and output stress as linear functions of the applied DC field [8]. M. Shibagaki et al. then developed a mathematical model of the static and dynamic characteristics to demonstrate the validity of the static model using a position-feedback-control method [9]. Y Li et al. modeled the nonlinear static relation between the displacement and the output force based on Hill’s muscle model [10]. K. Asaka et al. proposed an electromechanical model based on the electrochemical properties of PVC gels [6]. A comparison of the stored-energy approach, energy-balance approach, and Maxwell-stress-tensor approach to derive expressions for the electrically induced stress state in the dielectric material was made [11]. Based on the material properties of Young’s modulus and damping, Maxwell stress was applied to obtain the governing equations [12]. The Maxwell-stress-tensor approach was further applied in modelling the response of PVC actuators [13,14].

Most studies are focused on the responsive deformation of the actuators based on the Maxwell stress tensor under certain boundary conditions, especially volume incompressibility. On the other hand, the stress is not sufficient to describe a desired actuation effect in the planar spatial directions. There is not much work published that examines the response of PVC-gel-based actuators under combined mechanical and electrical loading.

In this work, a viscoelastic model based on Maxwell and Kelvin’s elements is developed. The purpose is to describe the responsive deformation of the actuator. The internal stress state accounts for combined electromechanical loading. The model parameters are tuned using data obtained from a unique experimental setup. Moreover, the memory effect and nonlinear effect of PVC gels and the capacitance effects in DC fields are considered, upon which the deformation due to creep, relaxation, and recovery is analytically obtained. The paper further intends to examine if the gel can be assumed to act similar when subjected to a mechanical force or subjected to an electrostatic force.

## 2. Materials

### 2.1. PVC-Gel Actuator

The PVC-gel-actuator system under consideration, as shown in Figure 1, consisted of electrical-insulating plates, electric conductors (foil and mesh), and a PVC-gel layer. The PVC gel was sandwiched between a stainless-steel mesh (32 mesh) as an anode and a foil as a cathode (Figure 1a). The grid size of the mesh was 0.568 mm (Figure 1b) and woven from stainless-steel wire of 0.268 mm in diameter. When the electric field was discharged, the PVC gels returned to their original shape.

### 2.2. PVC Gels

The fabrication of the PVC gel was started by dissolving the PVC powder into mixtures of THF and DBA in a beaker at room temperature. Then, the mixture was stirred evenly with a blender at 45 °C for 4 h. Afterwards, the mixture was cast in a Petri dish and dried in a vacuum drying oven at 0.06 MPa for 1 day. The final step included the last evaporation of the THF, which was carried out at room temperature for about three days to form a transparent PVC-gel membrane [15,16]. Aoki [17] characterized the residual THF concentration using the head space gas chromatography (HSGC) method. As a result, the PVC gel had a very low concentration of residual THF and the performance of the PVC gel was not affected by residual THF [2,18].

Typically, the PVC-gel film has a thickness of 1.0–1.2 mm. The PVC gels were cut to a suitable size for the actuator application.

PVC gels were made with a semicrystalline and 3D-crosslinking network PVC polymer main matrix (it is commonly commoditized PVC powder and the molecular weight is 48,000) and a plasticizer and dispersant of dibutyl adipate (DBA). The present PVC gels were prepared with 1:4 *w*/*w*% of the PVC and the DBA. The properties of tetrahydrofuran (THF) and DBA are shown in Table 1 and Table 2, respectively.

## 3. Experimental Setup

The PVC-gel actuator was installed in a dynamic mechanical analyzer from TA-Instruments (DMA Q850), which can impose a compressive mechanical force to the actuator sample (see Figure 2). It can provide 0.1 nm resolution over a 25 mm continuous range of travel for ultimate testing versatility and continuous forces from 0.1 mN to 18 N delivered by a low-mass motor. These mechanical compressive forces (F_m_) used in the experimental campaign were in the range from 3 N to 9 N. The electrical potential (U) between the anode and cathode was delivered by a conventional 30 V DC power supply (B&K precision 1671A) and amplified to a range from 400 V to 800 V. The anode and cathode were electrical insulated from the mechanical loading fixture of the DMA by two rigid polymer plates. The combined loading from the mechanical force (F_m_) delivered by the DMA and the electrostatic force (F_e_) was imposed by the electrical potential (U).

All tests are conducted at room temperature (23 °C). The loading of the actuator was performed by applying a constant mechanical force (F_m_) and two stepwise electrical potentials of magnitude (U). This created five loading phases, denoted Phase 1, Phase 2, Phase 3, Phase 4, Phase 5 (cf. Figure 3). In Phase 1, 3, and 5, only a mechanical compression force was exerted on the actuator, whereas in Phase 2 and 4, a combination of mechanical and electrical forces was imposed.

The test setup with the five loading phases was designed so that different viscoelastic responses were expected in the various loading phases.

Phase 1: viscoelastic creep due to the constant mechanical force (see lower graph in Figure 2).Phase 2 and 4: creep from the combined mechanical and electrical forces.Phase 3 and 5: continued creep for the mechanical force and viscoelastic recovery from removing the electrical potential.

A full factorial experimental campaign was conducted with three levels of electrical potentials U = [400 V, 600 V, 800 V] and four levels of mechanical forces F_m_ = [3 N, 5 N, 7 N, 9 N]. 

## 4. Model Formulation

In the model, the cross-influences of voltage or strain-dependent processes are given. The underlying model assumptions for the PVC-gel actuator are the following.

The response of the actuator under the given load conditions presented in Section 3 can be modelled with a 1D viscoelastic model.To explore the hypothesis made in the introduction that the PVC gel actuator responds similarly when exposed to a mechanical force or to an electrostatic force, it is, therefore, assumed that the model parameters can be derived from data regions dominated by the mechanical force.There is no consideration of anisotropic material behavior.The electrodes needed to apply an electric field are fully compliant and perfectly conductive.The applied charges are assumed to be homogeneously distributed over the electrodes.

The PVC gel in the actuator behaved like viscoelastic materials (cf. Figure 4). When an external force was applied, it caused strain and deformation in the material, while when the external force was removed, the material recovers from the deformation, which is time-dependent.

The mechanical force (F_m_) delivered using the DMA and the electrostatic force (F_e_) imposed by the electrical potential (U) may be expressed as in Equation (1) [19].
(1)$$F_{e}\left(t\right)=\frac{{\varepsilon \;S\;U}^{2}}{d{\left(t\right)}^{2}}=\frac{{\varepsilon \;S\;U}^{2}}{\;{\left(d_{0}-\delta \left(t\right)\right)}^{2}}\;\approx \frac{{\varepsilon \;S\;U}^{2}}{d_{0}{}^{2}}$$

In Equation (1), the permittivity of the gel is (ε) and the surface area of the capacitor plates (foil or metal mesh) is (S). The instant distance between the plates is (d) and may be expressed in terms of the initial plate distance (d_0_) and displacement (δ). The initial thickness of the gel is 1 mm (d_0_ = 1 mm) and the surface area of the stainless-steel mesh is 177 mm^2^ (S = 177 mm^2^). Notice that in Equation (1), small displacements compared to the initial plate distance are assumed small ($\delta \ll d_{0}$), which yields $\left(d_{0}-\delta \right)\approx d_{0}$.

The viscoelastic creep of the gel under the action of the constant mechanical force (F_m_) and a stepwise constant potential (U) may be expressed as in Equations (2) and (3), respectively. Furthermore, notice that the expressions in Equations (2) and (3) are rewritten in terms of the parameters $c_{1}=d_{0}/S$ and $c_{2}=\varepsilon /\left(d_{0}\right)=\left(\varepsilon_{0}\varepsilon_{r}\right)/\left(d_{0}\right)$ where the permittivity of vacuum is ($\varepsilon_{0}$) and the relative permittivity or the dielectric constant is denoted ($\varepsilon_{r}$). The dielectric constant is in this work set to ($\varepsilon_{r}=3000$) but may take values within a wide range 10–5000 and be heavily influenced by the loading frequency and plasticizer concentration [15].
(2)$$\begin{array}{c} \delta_{\mathrm{creep},\;\mathrm{Fm}}\left(t\right)\\ =\frac{F_{m}{\;d}_{0}}{S}\left(\underset{\mathrm{Maxwell}}{\underbrace{\frac{1}{\xi_{1}}+\frac{t}{\eta_{1}}}}+\underset{\mathrm{Kelvin}}{\underbrace{\frac{1}{\xi_{2}}\left(1+e^{\frac{-\xi_{2}\;t}{\eta_{2}}}\right)}}\right)\\ =c_{1}F_{m}\left(\frac{1}{\xi_{1}}+\frac{t}{\eta_{1}}+\frac{1}{\xi_{2}}\left(1+e^{\frac{-\xi_{2}\;t}{\eta_{2}}}\right)\right) \end{array}$$
(3)$$\begin{array}{c} \delta_{\mathrm{creep},\;\mathrm{Fe}}\left(t-t_{i}\right)\\ =\underset{F_{e}}{\underbrace{\left(\frac{{\varepsilon \;S\;U}^{2}}{d_{0}{}^{2}}\right)}}\frac{{\;d}_{0}}{S}\left(\underset{\mathrm{Maxwell}}{\underbrace{\frac{1}{\xi_{1}}+\frac{\left(t-t_{i}\right)}{\eta_{1}}}}+\underset{\mathrm{Kelvin}}{\underbrace{\frac{1}{\xi_{2}}\left(1+e^{\frac{-\xi_{2}\;\left(t-t_{i}\right)}{\eta_{2}}}\right)}}\right)\\ =c_{2}U^{2}\left(\frac{1}{\xi_{1}}+\frac{\left(t-t_{i}\right)}{\eta_{1}}+\frac{1}{\xi_{2}}\left(1+e^{\frac{-\xi_{2}\;\left(t-t_{i}\right)}{\eta_{2}}}\right)\right)\\ i\in \left[1,3\right],{\;t}_{i}<t<t_{i+1} \end{array}$$

The viscoelastic material parameters $\left(\xi_{1},{\;\eta }_{1},{\;\xi }_{2},{\;\eta }_{2}\right)$ are related to the stiffness of the PVC gel (cf. Figure 4). The creep induced by the mechanical force Equation (2) is taking place in the entire time domain as the force is constant. However, Equation (3) is only valid within the time domains specified which are Phase 2 and 4. In Phase 1, 3, and 5, ($\delta_{\mathrm{creep},\;\mathrm{Fe}}=0$) as (U=0) in these intervals. 

The total creep within Phase 2 and 4 will be the sum of Equations (2) and (3).

As the electrical force is removed for Phase 3 and 5, the material attempts to recover and this response is controlled by the parameters ($\xi_{1}$, $\xi_{2}$ and $\eta_{2}$), as described in Equations (4) and (5).
(4)$$\begin{array}{c} \delta_{\mathrm{creep},\;\mathrm{Fe}}\left(\Delta t_{U}\right)\\ =c_{2}U^{2}\left(\frac{1}{\xi_{1}}+\frac{\Delta t_{U}}{\eta_{1}}+\frac{1}{\xi_{2}}\left(1+e^{\frac{-\xi_{2}\;\Delta t_{U}}{\eta_{2}}}\right)\right) \end{array}$$
(5)$$\begin{array}{c} \delta_{\mathrm{recovery},\;\mathrm{Fe}}\left(t-t_{i+1}\right)\\ =\left(\delta_{\mathrm{creep},\;\mathrm{Fe}}\left(\Delta t_{U}\right)-\frac{c_{2}U^{2}}{\xi_{1}}\right)e^{\frac{-\xi_{2}\;\left(t-t_{i+1}\right)}{\eta_{2}}},\;\begin{array}{c} \;i\in \left[1,3\right]\\ \;t>t_{i+1} \end{array} \end{array}$$

In Equation (4), the magnitude of the creep at the end of Phase 2 (or Phase 4), due to the electrical potential (U) only, is determined. This is used as the initial stage for the recovery of Phase 3 (or Phase 5).

## 5. Model Predictions and Discussion

To determine the four viscoelastic parameters ($\xi_{1}$, $\eta_{1}$, $\xi_{2}$ and $\eta_{2}$) and, later, to investigate if these four parameters are universal for both mechanical and electrical loading, the only data that were dominated by the mechanical loading were used to determine the four parameters. These data regions are highlighted in Figure 5 as red shaded areas. In Phase 1, only mechanical force was applied to the sample. Moreover, it was assumed that at the end of Phase 3 and 5, the deformation due to recovery was minimal and the response of the PVC-gel actuator was governed by the creep from the mechanical force. 

Least square regression analysis was used in determining the four model parameters, which was carried out for each of the four studied load levels (cf. Table 3). It was seen that the model parameter showed dependency on the mechanical-load levels (F_m_). An example of the model prediction for a constant mechanical force of 3 N and no electrical potential applied is shown in Figure 5. It was observed that the model matched the data within the red shaded regions as expected. 

It was noted that the values for the model parameters in Table 3 are plotted with respect to the mechanical force (F_m_) and it can be seen that there exists a close-to-linear relation between them (cf. Figure 6). This further indicates that the four model parameters were not represented by a single value and, therefore, a linear function was used for each of them. 

The linear function parameters for each of the four viscoelastic parameters were similarly found using least square regression and their values are stated in Table 4.

By implementing the linear functions for $\xi_{1}$, $\eta_{1}$, $\xi_{2}$ and $\eta_{2}$ in Equations (2) and (3), the model predictions can be visualized against the entire data set (cf. Figure 6). The model predictions are based on the scalar (dielectric constant, $\varepsilon_{r}$), which controls the model predictability in Phase 2 and 4 but also, to some extent, the two recovery regions Phase 3 and 5.

In Figure 7, the test data generally show that increased mechanical force resulted in increased displacement, which was equal to a larger compaction of the PVC gel. This was also captured by the model, as long as the scatter in the experimental data is considered. Notice that the model assumes ($\varepsilon_{r}=3000$). Furthermore, the applied electrical potentials of 400 V, 600 V, and 800 V correspond to electrostatic forces (F_e_) of 0.8 N, 1.7 N, and 3.0 N, respectively.

As the electrical potential was applied in Phase 2 and 4, the data indicate a negative correlation between the displacement step associated with the electro-static force and the applied mechanical force. This observation is further supported by the fact that viscoelastic model parameters increased in magnitude with increasing mechanical force (see Figure 6).

However, in general, the model seems not to be able to capture the displacements and its trends well in Phase 2 and 4. Therefore, attention is drawn to the instant displacement change ($\Delta \delta_{\xi_{1}}$) in Phase 2 and 4, in which its magnitude was controlled by U^2^ and stiffness parameter ($\xi_{1}$). The displacement changes were evaluated from test data in Phase 2 and are graphically shown in Figure 8 for different levels of U. In the right-side graph of Figure 8, the data indicate an increasing displacement change with increasing electrical potential (U). Moreover, model predictions using scaled model parameters to match the experimental data are shown with grey dashed lines. The scaling factor needed was around 10, which was a good match for electrical potentials of 400 V and 600 V but not for 800 V.

This further means that the model parameters ($\xi_{1}$, $\eta_{1}$, $\xi_{2}$ and $\eta_{2}$), determined based on data from mechanical-force-rich regions in the data set, cannot be used to predict the displacement when an electrostatic force is applied. The viscoelastic behavior of the actuator is simply dependent on the type of applied force (either mechanical or electro-static).

In addition, the data in Figure 8 indicate that separate model parameters for Phase 2 and 4 are required and assuming linear functions for them will not be sufficient.

For the data based on U = 800 V, the model predictions of the displacement in Phase-2 were better, but there was a relatively visible error of a cross trend between the experimental data and the model in Phase 4. 

The displacements due to the mechanical load tended to increase with increasing step input in U (see Figure 8). The displacement for U = 800 V was 2–3 times of that for U = 400 V. It indicated that there was more time for recovery due to the increased displacement and the divergence of the modelling error due to the longer recovery time. The error can be minimized by picking the natural frequency of the system [12].

## 6. Conclusions

A viscoelastic model based on a combined Maxwell and Kelvin–Voigt model was developed to describe the responsive deformation of the actuator under combined mechanical and electrical loading. Least square regression analysis was used in determining the four model parameters. The results showed a great correlation between the model and the experiment outputs, indicating a negative correlation between the displacement step associated with the electrostatic force and the applied mechanical force. The strain initially reached a value under the step input voltage and then slowly creeped toward another point while the input was held constant. Due to the memory effect and nonlinear effect of PVC gels and the capacitance effects in DC fields, there was more time for recovery.

The proposed model for the actuator under investigation can be used to describe the responsive deformation due to a mechanical force only. Further work and understanding of the viscoelastic behavior of PVC gels needs to be achieved before the model is able to describe the deformation under combined mechanical and electric forces.

## Figures and Tables

**Figure 1 materials-16-01183-f001:**
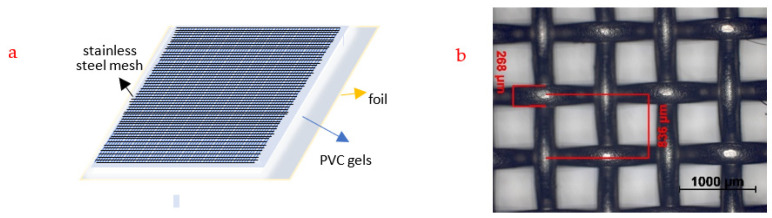
(**a**) Schematics of the PVC-gel actuator tested. (**b**) stainless-steel mesh.

**Figure 2 materials-16-01183-f002:**
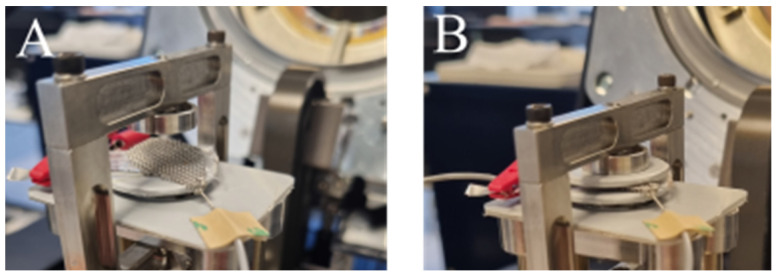
Experiment schematic of PVC-gel actuator. (**A**) PVC-gel actuator loaded with only electrostatic force. (**B**) PVC-gel actuator loaded with both mechanical force and electrostatic force.

**Figure 3 materials-16-01183-f003:**
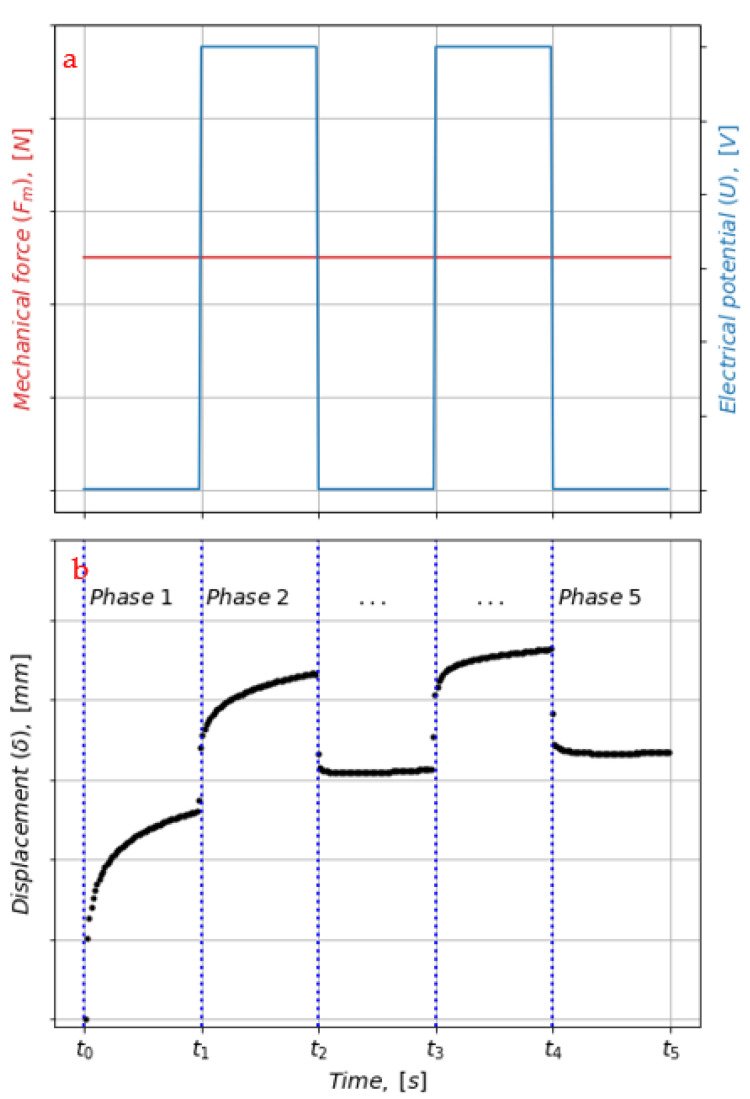
Test conditions illustrating that the gel actuator was subjected to a variety of combinations of constant mechanical force (F_m_) and steps of electrical potentials (U). (**a**) Mechanical force (F_m_) and steps of electrical potentials (U). (**b**) Displacement (δ) of PVC-gel actuator loaded with both mechanical force and electrostatic force.

**Figure 4 materials-16-01183-f004:**
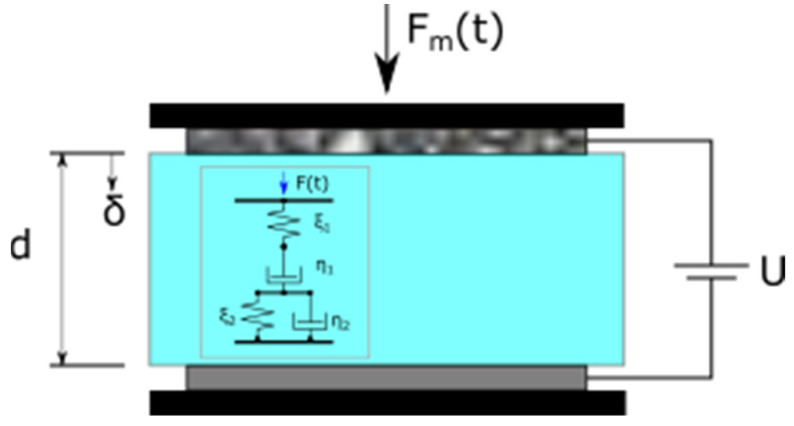
The actuator is considered to be modelled where the PVC gel is considered to be viscoelastic.

**Figure 5 materials-16-01183-f005:**
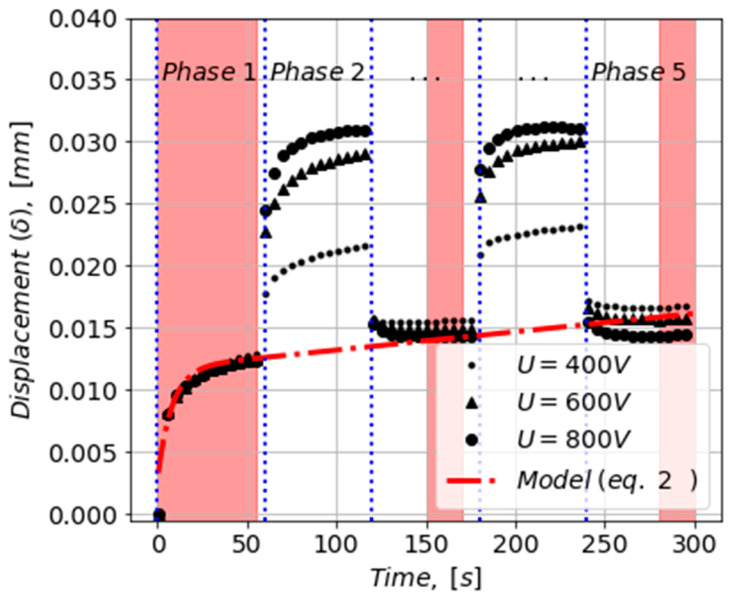
Data collected for F_m_ = 3 N and for three magnitudes of U.

**Figure 6 materials-16-01183-f006:**
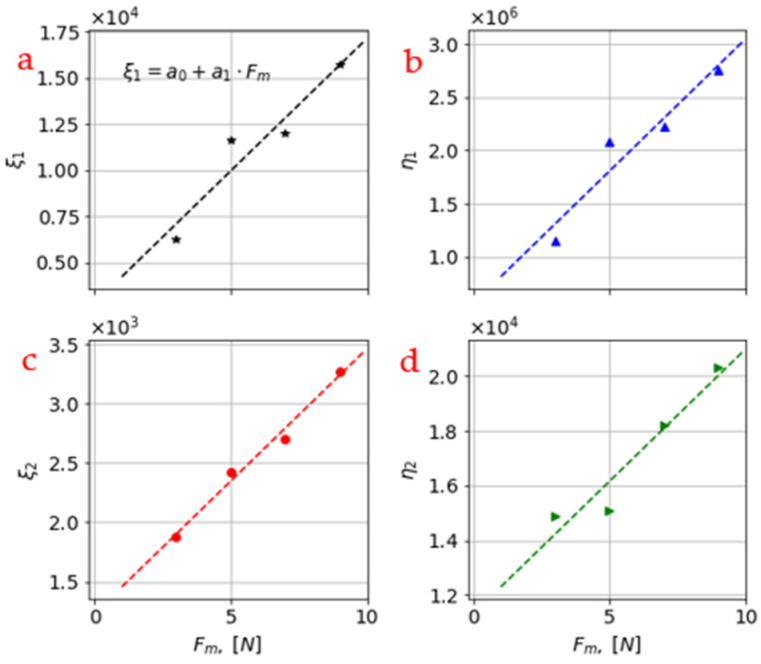
The individual parameters for each mechanical-loading level are plotted here to generate model parameters that are linearly dependent on the mechanical force (F_m_). F_m_ is given as 3 N, 5 N, 7 N, and 9 N. (**a**) $\xi_{1}$. (**b**) $\eta_{1}$. (**c**) $\xi_{2}$. (**d**) $\eta_{2}$.

**Figure 7 materials-16-01183-f007:**
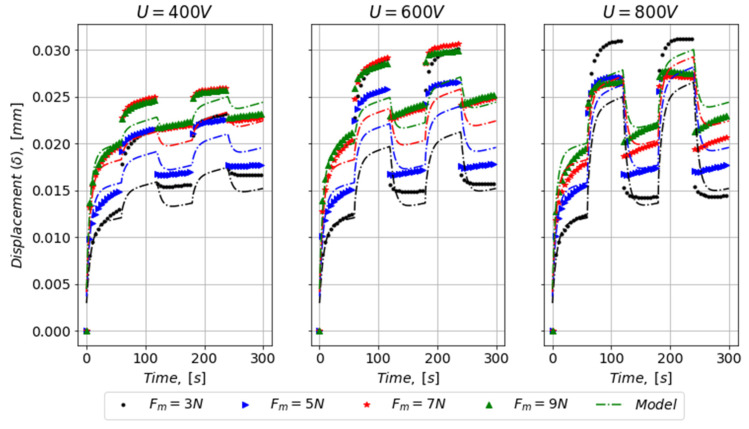
Model predictions against the entire data set.

**Figure 8 materials-16-01183-f008:**
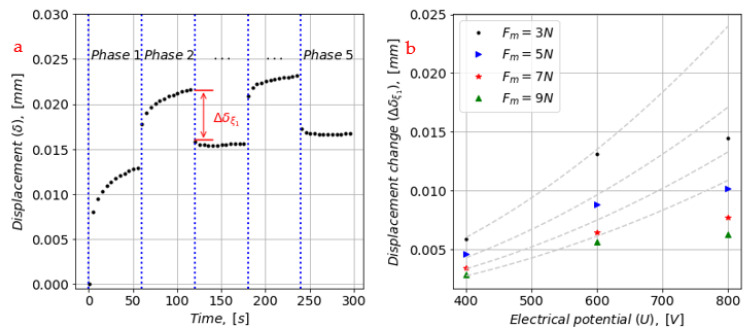
(**a**) The instant change in displacement ($\Delta \delta_{\xi_{1}}$) in Phase 2. (**b**) The data for all load combinations of F_m_ and U. Model predictions using scaled model parameters are shown with grey dashed lines (scaling factor of 10 is used to match data for U = 400 V).

**Table 1 materials-16-01183-t001:** Properties of THF.

Property	Boiling Point (°C)	Density (g/cm3)	Flashpoint (°C)	Purity (%)	Viscosity (mPa.s)
Value	65	0.89	−21	99	0.48

**Table 2 materials-16-01183-t002:** Properties of DBA.

Property	Assay (%)	Refractive Index (n20/D)	Boiling Point (°C)	Density (g/mL)	Melting Point (°C)
Value	96	1.436	305	0.962	−32

**Table 3 materials-16-01183-t003:** Model constants when fitted to data from the individual load levels.

	$F_{m}=3\;N$	$F_{m}=5\;N$	$F_{m}=7\;N$	$F_{m}=9\;N$
$\xi_{1}\;[N/{\mathrm{mm}}^{2}]$	6.28 × 10^3^	1.16 × 10^4^	1.20 × 10^4^	1.57 × 10^4^
$\eta_{1}\;[N\;s/{\mathrm{mm}}^{2}]$	1.15 × 10^6^	2.08 × 10^6^	2.23 × 10^6^	2.76 × 10^6^
$\xi_{2}\;[N/{\mathrm{mm}}^{2}]$	1.88 × 10^3^	2.42 × 10^3^	2.70 × 10^3^	3.27 × 10^3^
$\eta_{2}\;[N\;s/{\mathrm{mm}}^{2}]$	1.49 × 10^4^	1.51 × 10^4^	1.82 × 10^4^	2.03 × 10^4^

**Table 4 materials-16-01183-t004:** Linear model parameters for $\xi_{1}$, $\eta_{1}$, $\xi_{2}$ and $\eta_{2}$. Each model parameter is given as $\xi_{1}=a_{0}+a_{1}F_{m}$, $\eta_{1}=a_{0}+a_{1}F_{m}$ etc.

	$a_{0}$	$a_{1}$
$\xi_{1}\;[N/{\mathrm{mm}}^{2}]$	2797	1433
$\eta_{1}\;[N\;s/{\mathrm{mm}}^{2}]$	561,000	249,000
$\xi_{2}\;[N/{\mathrm{mm}}^{2}]$	1232	223
$\eta_{2}\;[N\;s/{\mathrm{mm}}^{2}]$	11,335	965

## Data Availability

Not applicable.

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
