# Peer review of "Investigation of a Simple Viscoelastic Model for a PVC-Gel Actuator under Combined Mechanical and Electrical Loading"

_materials, 2023, doi:10.3390/ma16031183_

Round 1

Reviewer 1 Report

Attached 

Author Response

Point 1: Can you please enhance the view of the abstract section by adding some of your real results and findings, values to show clear evidence about your results

Response 1: Thank you. I have revised as the advises.

Point 2: What are the used parameters of calculating the mesh?

Response 2: The PVC gel is sandwiched between a stainless steel mesh (32 mesh) as an anode and a foil as cathode (Fig.1a). The grid size of the mesh is 0.568 mm (Fig.1b) and woven from a stainless steel wire of 0.268 mm in diameter.

Point 3: The resolution of Fig 2 is not so clear, may you insert a better resolution

Response 3: Thank you. I have revised as the advises.

Point 4: In Table 2, did you measure the refractive index or is it taken from a source (in this case please mention the reference)

Response 4: Thank you. We didn’t measure the refractive. We get it from instruction of DBA.

Point 5: In Fig 5 can you add the response and recovery time of these curves, many references can be found here Nanomaterials | Free Full-Text | Nanostructured MoS2 and WS2 Photoresponses under Gas Stmuli (mdpi.com) Then you have to plot a curve between the obtained values of each sample here and the response time and another curve for the other case of the recovery time

Response 5: we are not clear on what the reviewer means by his/her comment. However, the objective with Fig-5 is to illustrate the three data sets in black agree within the red shaded regions in which is dominated by creep due the mechanical force applied. Therefore the data within these regions are used to determine the model parameters. We hope this may have clarified our intentions with this illustration.

Ref: [1]Basyooni, M.A.; Zaki, S.E.; Alfryyan, N.; Tihtih, M.; Eker, Y.R.; Attia, G.F.; Yılmaz, M.; AteÅŸ, Åž.; Shaban, M. Nanostructured MoS2 and WS2 Photoresponses under Gas Stimuli. Nanomaterials 202212, 3585. https://doi.org/10.3390/nano12203585

Point 6: The conclusion part must be enhanced, you may keep only one paragraph summarizing your results in a better way.

Response 6: Thank you. I have revised as the advises.

A visco-elastic model based on combined Maxwell and Kelvin-Voigt model is developed to describe the responsive deformation of the actuator under combined mechanical and electrical loading. Least square regression analysis is used in determining the four model parameters. The results show a great correlation between the model and the experiment outputs, indicating a negative correlation between the displacement step associated to the electrostatic force and the applied mechanical force. The strain would initially reach a value under the step input voltage and then slowly creep toward another point while the input is held constant. Due to the memory effect and nonlinear effect of PVC gels and the capacitance effects in DC fields, there is more time for recovery.

The proposed model for the actuator under investigation can be used to describe the responsive deformation due to a mechanical force only. Further work and under-standing of the visco-elastic behavior of the PVC-gel needs to be achieved before the model is able to describe the deformation under combined mechanical- and electric forces.

Point 7: Do you think that your mechanical measurement system is better or using a Nanondenation system? and why?

Response 7:  I am unfamiliar with Nanondenation system. Nanondenation system maybe more useful in getting the deformation mechanism of PVC gel actuator. Mechanical measurement could get the relationship between deformation and electric and mechanical loading easily.

Point 8: The captions of the figures are not the same, some times are red (Fig 5) and white (fg 1). It must be the same color and size

Response 8: Thank you. I have revised as the advises.

Reviewer 2 Report

The authors developed a model incorporating viscoelasticity to describe the responsive deformation of PVC gel actuators. However, there were many elementary errors in the text, and I requested a major revision.

Assumes a dielectric constant of 3000, but does not provide sufficient evidence for this assumption.

How is the surface area of an electrode consisting of a mesh calculated? If it is just sandwiched, should the surface area change with pressure?

The authors have described PVC gels, the film has a thickness of 1.0-1.5mm(line82). On the other hand, it is stated that the electrostatic force is inversely proportional to the square of the film thickness. It is thought that the electrostatic force generated varies greatly depending on the film thickness, but how is this standardized? Also, the description of the initial thickness of line 151 as 1 mm conflicts with that of line 82.

For readers unfamiliar with PVC gel actuators, some of the wording is difficult to understand. For example, it does not describe what PVC stands for. It isn't easy to understand what kind of dispersant is used for the gel. ( It should also be noted that DBA is a dispersant and a plasticizer.)

What do Tables 1 and 2 want to show? I thought that refractive index, density, flash point, and other values do not seem relevant to this paper. If you're going to state that DBA acts as a dispersant of PVA gel, you'd also need a melting point.

THF is removed by decompression drying, but at what pressure? Why is the THF removed by decompression drying and then allowed to stand at room temperature for 3 days? If the THF is completely out, then 3 days of static at room temperature is not necessary, and if it's not dry enough, then decompression drying should be prolonged. Also, no evidence has been presented to show that THF has been completely removed

Line 66: Two consecutive periods.

Line 153: mm2 (superscript)

Some mechanical force (Fm) m's are not subscripted.

Put a space between units and numbers.

Unify your expressions. (e.g.: Eq. Eqn. Eq- )

The text in the figure is difficult to read.

Author Response

Point 1: Assumes a dielectric constant of 3000, but does not provide sufficient evidence for this assumption.

Response 1: The dielectric constant of different PVC gel was tested in the reference. We elected the PVC gel in the reference more same with our experiment PVC gel in the weight ratio of PVC to DBA and the molecular weights. And We also have tried our best to test in our model.

Ref: [1] Shin E J, Park W H, Kim S Y. Fabrication of a high-performance bending actuator made with a PVC gel[J]. Applied Sciences, 2018, 8(8): 1284.

Point 2: How is the surface area of an electrode consisting of a mesh calculated? If it is just sandwiched, should the surface area change with pressure?

Response 2: The surface area of the actuator is the surface area of the mesh. Because the PVC gel is a little bigger than the mesh.

Point 3: The authors have described PVC gels, the film has a thickness of 1.0-1.5 mm(line82). On the other hand, it is stated that the electrostatic force is inversely proportional to the square of the film thickness. It is thought that the electrostatic force generated varies greatly depending on the film thickness, but how is this standardized? Also, the description of the initial thickness of line 151 as 1 mm conflicts with that of line 82.

Response 3: There is a written error. The film has a thickness of 1.0-1.2 mm. In the experiment, we used the same film cut from the same place of the big film in order to get the uniformity film as 1 mm.

Point 4: For readers unfamiliar with PVC gel actuators, some of the wording is difficult to understand. For example, it does not describe what PVC stands for. It isn't easy to understand what kind of dispersant is used for the gel. (It should also be noted that DBA is a dispersant and a plasticizer.)

Response 4: We will add the information on section2.2.

PVC gels were made with a semicrystalline and 3D crosslinking network PVC polymer main matrix (it is commonly commoditized PVC powder and molecular weight is 48000) and a plasticizer and dispersant of dibutyl adipate (DBA).

Point 5: What do Tables 1 and 2 want to show? I thought that refractive index, density, flash point, and other values do not seem relevant to this paper. If you're going to state that DBA acts as a dispersant of PVA gel, you'd also need a melting point.

Response 5:The properties of the PVC gel is different with different DBA. The properties of THF results in different prepared processing. And we have added the melting point.

Table 2 Properties of DBA

Property

Assay

(%)

Refractive index

(n20/D)

Boiling point

(°C)

Density

(g/ml)

melting point

(°C)

Value

96

1.436

305

0.962

-32

Point 6: THF is removed by decompression drying, but at what pressure? Why is the THF removed by decompression drying and then allowed to stand at room temperature for 3 days? If the THF is completely out, then 3 days of static at room temperature is not necessary, and if it's not dry enough, then decompression drying should be prolonged. Also, no evidence has been presented to show that THF has been completely removed

Response 5: In order to remove THF as soon as possible, the THF is removed by decompression drying at 0.06Mpa. In most experiment of references, the THF is removed by decompression drying for 2 days or at room temperature for 4-5 days of static [1][2]. To evaporate the THF fully, we put the gel at room temperature for 3 days of static after decompression drying.

To determine the residual THF concentration in the PVC gel, Aoki characterized the residual THF concentration by head space gas chromatography (HSGC) method. As a result, the PVC gel has very low concentration of residual THF and the performance of PVC gel is not affected by residual THF [3][4].

Ref:[1] Hong X , Takasaki M , Hirai T . Actuation mechanism of plasticized PVC by electric field[J]. Sensors & Actuators A Physical, 2010, 157(2):307-312.

[2] Shin E J, Park W H, Kim S Y. Fabrication of a high-performance bending actuator made with a PVC gel[J]. Applied Sciences, 2018, 8(8): 1284.

[3] Liu Z , Liu Y D , Shi Q , et al. Electroactive dielectric polymer gels as new-generation soft actuators: a review[J]. Journal of Materials Science, 2021:1-21.

[4] Yuji, Aoki. A Preparation Method of Thermoreversible Poly(vinyl chloride) Gels[J]. Macromolecules, 2001.

Point 7: Line 66: Two consecutive periods.

Response 7:  Thank you. I have revised as the advises.

Point 8: Line 153: mm2 (superscript)

Response 8:  Thank you. I have revised as the advises.

Point 9: Some mechanical force (Fm) m's are not subscripted.

Response 9:  Thank you. I have revised as the advises.

Point 10: Put a space between units and numbers. Unify your expressions. (e.g.: Eq. Eqn. Eq- )

Response 10:  Thank you. I have revised as the advises.

Point 11: The text in the figure is difficult to read.

Response 11:  Thank you. I have revised as the advises.

Round 2

Reviewer 1 Report

The authors did the required corrections. 

Reviewer 2 Report

The authors have made sufficient corrections to the content to make it suitable for publication in Materials.